# Association of APOE Serum Levels and *APOE* ε2, ε3, and ε4 Alleles with Optic Neuritis

**DOI:** 10.3390/genes13071188

**Published:** 2022-07-01

**Authors:** Liucija Momkute, Alvita Vilkeviciute, Greta Gedvilaite, Gabriele Dubinskaite, Loresa Kriauciuniene, Rasa Liutkeviciene

**Affiliations:** 1Medical Academy, Lithuanian University of Health Sciences, LT-50161 Kaunas, Lithuania; liucija.momkute@stud.lsmuni.lt (L.M.); gabriele.dubinskaite@stud.lsmuni.lt (G.D.); 2Ophthalmology Laboratory, Neuroscience Institute, Lithuanian University of Health Sciences, Medical Academy, LT-50161 Kaunas, Lithuania; alvita.vilkeviciute@lsmuni.lt (A.V.); greta.gedvilaite@lsmuni.lt (G.G.); loresa.kriauciuniene@lsmuni.lt (L.K.)

**Keywords:** optic neuritis (ON), APOE, ε3/ε3

## Abstract

Optical neuritis (ON), otherwise known as optical nerve damage, is a term used to describe various environmental and body conditions that lead to optic nerve dysfunction. Neurologists are well aware of conditions that cause optic neuropathy, such as trauma, infections, malnutrition, and various toxins. As optic neuritis is a multifactorial demyelinating or infectious process, genetic predisposition may also influence the progression of optic neuritis. This study aimed to evaluate the association of ON (with and without multiple sclerosis) with *APOE* alleles and APOE serum levels. We found that the *APOE* ε3/ε3 genotype was statistically less common in the ON group of males than in the control group (*p* = 0.045). Moreover, the *APOE* ε3/ε3 genotype had a 3.7-fold increase in the odds of ON development in males (OR = 3.698; CI: 1.503–9.095; *p* = 0.004). In contrast, the *APOE* ε3/ε4 genotype had a 4.1-fold decrease in the odds of ON development in males (OR = 0.242; CI: 0.083–0.704; *p* = 0.009). APOE serum levels were statistically significantly higher in the ON group than in the control group (*p* = 0.042). The *APOE* ε3/ε3 genotype may increase males’ risk of developing ON, while the ε3/ε4 genotype may reduce males’ risk of developing ON.

## 1. Introduction

Optic neuritis (ON) is an acute inflammatory disorder demyelinating the optic nerve. The isolated optic neuritis characteristics include unilateral, acute, and painful vision loss without systemic or other neurological symptoms [1]. In most cases, the etiology of the disease may be unknown, so it is called idiopathic ON [1]. The mechanism responsible for acute ON has not been definitively identified, but it is believed that an autoimmune reaction damages the myelin sheath that encompasses the neurons within the optic nerve [2]. When a disturbed nerve impulse causes an inflammatory response, type IV hyperallergic reactions occur. Activated peripheral T lymphocytes begin to secrete cytokines (inflammatory mediators), which easily cross the blood–cerebral protective barrier and cause myelin rupture. ON disease occurs when inflammatory mediators with T lymphocytes reach the central nervous system [3]. When the initial ON episode occurs, the patient experiences decreased color contrast sensitivity (depending on the period, blue and yellow in acute cases, and red and green in non-acute cases), photopsia (“flashes of light”), or impaired depth sensation in moving objects (Pulfrich phenomenon) [4,5]. Further tests are then applied to confirm the diagnosis of the disease. For evaluation, routine blood and magnetic resonance imaging, cerebrospinal fluid testing, and visual evoked potential tests are performed [1].

There is a close association between optic neuritis and multiple sclerosis (MS). MS is a chronic inflammatory autoimmune demyelinating disease that affects the central nervous system. Its mechanism of action is linked to the immune system; it destroys myelin and forms scar tissue, which ultimately results in a disruption of nerve impulse transmission [6]. Studies have shown that ON occurs in approximately 70% of patients with MS [7]. Swelling of the optic disc may be observed during ophthalmoscopy. The retina is usually unchanged, but the resulting peripheral lining may be associated with an increased risk of multiple sclerosis. Therefore, optic neuritis may also be considered one of the manifestations of multiple sclerosis [8].

Optic neuritis is considered to be a multifactorial demyelinating process. The causes of the disease have also been examined at the genetic level, but they are not exactly known yet. APOE consists of 299 amino acids encoded in chromosome 19q13. The gene is involved in transporting fat molecules associated with low-density lipoproteins (LDLs) and very low-density lipoproteins (VLDLs). Studies have shown that the *APOE* gene is essential in the healing process of various CNSs and that this function is impaired when the protein exists in the ε4 isoform (encoded by the ε4 allele). Thus, we aimed to investigate the impact of *APOE* on ON development. A total of three isoforms are known to exist in the *APOE* gene. APOE2, APOE3, and APOE4 are the products of three alleles (ε2, ε3, and ε4) at a single gene locus. The ε3 isoform is considered usual. Higher frequencies of specific alleles are observed in certain diseases, e.g., ε4 in Alzheimer’s disease [9,10,11,12,13]. APOE2 has cysteine in both positions, APOE3 has cysteine in 112 positions and arginine in 158 positions, and APOE4 has arginine in both positions. These are the products of three alleles (ε2, ε3, and ε4) in a single gene locus. There are six genotypes in the general population: three homozygous (*APOE* ε2/ε2, *APOE* ε3/ε3, and *APOE* ε4/ε4) and three heterozygous (*APOE* ε2/ε3, *APOE* ε2/ε4, and 3 A). The ε3 allele is the most common and is found in 77% of the general population. Meanwhile, the ε2 allele is found in 8% of the general population, and the ε4 allele is found in 15% of the general population. However, the frequency of *APOE* alleles varies between different ethnic groups [9,10].

In the periphery, APOE is synthesized by the liver, macrophages, and astrocytes, which are transmitted to neurons through LDL receptors. APOE2 binds to LDL less strongly than APOE3 and APOE4. Consequently, APOE2 carries lipids less efficiently, and ε2 is homozygous for the risk of type III hyperlipoproteinemia. APOE4 binds to lipoprotein particles and, therefore, is associated with an increased risk of hypercholesterolemia and atherosclerosis, a faster progression of HIV infection, and accelerated telomere shortening. The ε4 allele is a vital genetic risk factor for late-onset Alzheimer’s disease and neurodegenerative diseases, such as cerebral amyloid angiopathy, Lewy cell dementia, and MS. APOE2 has been associated with white matter loss in the brain, traumatic brain healing, and hemorrhagic and ischemic stroke [11,12,13,14,15].

The apolipoprotein E gene has also been linked to the development of another pathology, perpetual macular degeneration. Studies have confirmed that the ε4 allele provides protective care by reducing the risk of early-onset macular degeneration (AMD) and late AMD [16,17] in individuals with a family history of AMD. Individuals with the ε4 allele have a lower risk of developing the disorder than those with the ε2 allele, whose risk becomes higher for the development of AMD [18,19].

We aimed to find an association between optic neuritis with and without multiple sclerosis and the *APOE* allele and APOE serum levels.

## 2. Materials and Methods

The permission for this study was obtained from the Ethics Committee for Biomedical Research (No. BE–2–102). This study was carried out in the Department of Ophthalmology and at the Institute of Neurosciences, the Lithuanian University of Health Sciences (LUHS).

From 1 January 2012 to 1 February 2022, patients with acute ON from the Department of Ophthalmology were enrolled. The study participants were 87 subjects with optic neuritis and 583 healthy control subjects.

The criteria for the inclusion of patients with optic neuritis were as follows:The first occurrence of an acute ON episode;Patient’s consent to participate in the study;Consultation with a neurologist;Neurological examination if multiple sclerosis has been diagnosed according to McDonald’s (2005) criteria.

The exclusion criteria for study patients were as follows:Other diseases related to the optic nerve;Systemic diseases, such as diabetes, chronic infectious diseases, and oncological diseases after transplantation.

Other criteria for the inclusion of the population not suffering from an attack of optic neuritis were as follows:A positive diagnosis of other ocular diseases (after ophthalmological examination);Exclusion of other chronic/acute systemic diseases;Consent to participate in the study.

The exclusion criteria for healthy subjects were as follows:A negative diagnosis of ocular disease or visual impairment;Therapeutic disorders of other systems.

The average age of the control group and patients with optic neuritis was statistically significantly different (*p* < 0.05). Thus, a further analysis was performed and adjusted by age and gender. The demographic data of the subjects are shown in Table 1.

DNA was isolated from peripheral blood leukocytes using silica gel columns from a Thermo Scientific GeneJET Genomic DNA Purification Kit. This kit allows for the rapid and efficient extraction of high-quality DNA and is suitable for further RT-PCR assays. Two single-nucleotide polymorphisms of the *APOE* gene (rs429358 and rs7412) were evaluated using the genotype sets C___3084793_20 and C____904973_10 (Applied Biosystems, Foster City, CA, USA). The program started by heating for 10 min at 95 °C, then heating for 15 s to 95 °C for another 40 heating cycles, and finally raising the temperature to 60 °C for another 1 min. A Human Apo E (AD2) ELISA Kit was used to assess serum APOE levels.

Statistical analysis was performed using IBM SPSS Statistics 27.0. The qualitative characteristics of the subjects are expressed in absolute numbers and percentages in parentheses. The hypothesis about the normal distribution of the measured trait values was tested using the Kolmogorov–Smirnov and Shapiro–Wolf tests. The following descriptive statistical characteristics used for the study did not meet the criteria for a normal distribution: median and IQR. The Mann–Whitney U test was used to assess the difference between the two independent groups. The distributions of *APOE* genotypes and alleles in the ON and control groups were assessed using χ² and Fisher unilateral criteria calculations. Moreover, Hardy–Weinberg’s equilibrium (HWE) was performed in the control group. Subjects were divided into two groups (≤35 years old and >35 years old) based on the median age of all subjects to evaluate the distributions of *APOE* genotypes and alleles within younger and older subjects. Binary logistic regression was used to assess the influence of genotypes on ON development, indicating an OD of 95% confidence interval (CI). The differences were considered statistically significant when *p* < 0.05.

## 3. Results

*APOE* rs429358 and rs7412 genotyping was performed to evaluate the differences between the ON and control groups. Unfortunately, no statistically significant differences were found. The results are presented in Table 2.

Binary logistic regression was performed to evaluate the impact of *APOE* on the development of ON. No statistically significant results were found (Table 3).

The *APOE* rs429358 and rs7412 polymorphisms were evaluated in the ON and control groups according to the gender of the subjects. The *APOE* ε3/ε3 genotype was statistically significantly less common in the ON group of males than in the control group of males (48.4% vs. 66.4%, *p* = 0.045). The results are shown in Table 4.

A binary logistic regression analysis was performed in the ON and control groups to assess the effects of APOE genotypes on the incidence of ON by gender. We found that the *APOE* ε3/ε3 genotype is associated with a 3.7-fold increase in the odds of ON development in males (OR = 3.698; CI: 1.503–9.095; *p* = 0.004). In contrast, the *APOE* ε3/ε4 genotype is associated with a 4.1-fold decrease in the odds of ON development in males (OR = 0.242; CI: 0.083–0.704; *p* = 0.009). The results are shown in Table 5.

This study aimed to compare the genotypes and allele frequencies of the *APOE* in different groups of ON: ON without multiple sclerosis vs. the control group and ON with multiple sclerosis vs. the control group. However, no statistically significant differences were found between the groups. The data of the subjects are shown in Table 6.

A binary logistic regression analysis was performed in patients with ON without MS and with MS. No statistically significant differences were found between patients with ON without MS and patients with ON with MS. The results are shown in Table 7.

Serum APOE levels were measured in the ON (*n* = 20) and healthy subject (*n* = 20) groups. An analysis was performed using the Mann–Whitney U test. A statistically significant difference was found between the groups (median (IQR): 136.65 (75.71) vs. 107.89 (25.14), *p* = 0.042). APOE serum levels were statistically significantly higher in the ON group than in the control group. The results are shown in Figure 1.

## 4. Discussion

This study examined single-nucleotide polymorphisms rs429358 and rs7412 in the *APOE* gene in 87 patients with ON and 583 healthy subjects (without other ophthalmological and systemic diseases). HWE was performed in the control group (*n* = 583). The *APOE* rs429358 and rs7412 genotype distributions were consistent with HWE (*p =* 0.828 and 0.233, respectively). The frequency of genotypes and alleles was analyzed, but there were no statistically significant differences between the ON and control groups. A further analysis of SNPs was performed to evaluate the impact of age on ON development; unfortunately, it did not reveal any statistically significant differences. In contrast, statistically significant differences were found when the analysis was based on gender (*p* < 0.05). The *APOE* ε3/ε3 genotype was statistically significantly less frequent in males with ON than in control group males (*p* = 0.045). Moreover, the *APOE* ε3/ε3 genotype was associated with a 3.7-fold increase in the odds of ON development in males (*p* = 0.004). In contrast, the *APOE* ε3/ε4 genotype was associated with a 4.1-fold decrease in the odds of ON development in males (*p* = 0.009).

Many studies have analyzed the association of *APOE* with MS, but the data are contradictory. Many studies support the notion that the *APOE* ε4 allele is not associated with a higher risk of developing MS [20,21,22,23,24,25,26,27,28,29,30,31,32,33,34,35,36,37,38,39]. However, other studies suggest the opposite, stating that patients carrying the *APOE* ε4 allele may have decreased neuronal regeneration, leading to a faster progression of MS [27,40,41,42,43]. Moreover, a study conducted by Oliveri and co-authors found that the *APOE* gene−491A/T polymorphism is associated with severe cognitive impairment in patients with MS with a homozygous AA genotype [34]. Other researchers have found that *APOE* ε4 carriers are more likely to develop a more severe form of MS, while *APOE* ε2 carriers are more likely to develop a milder form of MS [44,45]. Julian and co-authors conducted a study to elucidate the association of the *APOE* gene with depression, which often occurs in patients with MS. Their results concluded that the *APOE* ε2 allele shows a protective effect by reducing the incidence of depressive disorders [46].

Only one study conducted by Pinholt et al. analyzed the association of *APOE* gene single-nucleotide polymorphisms with optic neuritis and multiple sclerosis. The study concluded that the *APOE* gene does not affect the development of either ON or MS [31]. The study looked for associations between the ε4 allele of the *APOE* gene and ON or MS. No statistically significant difference in alleles was found between genders. Researchers concluded that Apo E genotypes do not influence the development of MS or ON, but the Apo epsilon4 allele seems to predispose carriers with MS to a faster progression of disease [31].

However, Lazo et al. (a Mexican study) (abstract) examined 58 patients with ON and found no association between *APOE* and ON development [47]. In contrast, we found that the *APOE* ε3/ε3 genotype may reduce males’ risk of developing ON. Moreover, we found that APOE serum levels were statistically significantly higher in the ON group than in the control group (median (IQR): 136.65 (75.71) vs. 107.89 (25.14), *p* = 0.042). In comparison, a study conducted by Tamam et al. revealed that male patients with MS were likely to have lower serum APOE levels and that the most common *APOE* genotype in patients with MS was ε3/ε3 (82.0%). Male patients with MS were significantly more likely to have ε4, and at baseline, the disease duration was shorter, the Expanded Disability Status Scale (EDSS) scores were higher, the serum total APOE levels were lower, and the PI was significantly higher. The MS onset age, clinical types, EDSS scores, and PI had no significant correlation with the ε4 allele. Visual onset and sensory onset are good prognostic factors. There were no patients with visual onset and only a few patients with sensory onset in the ε4-positive group [48]. No statistically significant associations were found between APOE and ON with or without MS, but further and larger scale studies are needed to evaluate the associations of *APOE* SNPs and APOE serum levels with ON (with and without MS). Moreover, the links between *APOE* gene polymorphisms, Alzheimer’s disease, and cardiovascular disease have been extensively studied. There is scientific evidence that the ε4 allele of this gene is involved in the pathogenesis mechanism and influences the development of the mentioned diseases [9,10,11,12,13,14,15]. Kulminski et al. found that the *APOE* gene, along with five other single-nucleotide polymorphisms, influenced the pathogenesis of Alzheimer’s disease [49].

The strengths and limitations of our study are as follows: this is the first study to examine the relationship between APOE serum levels and *APOE* ε2, ε3, and ε4 alleles with ON and MS in the Baltic states, and this is the first study to divide patients into two groups, namely, patients that only had isolated optic neuritis and patients that had optic neuritis and multiple sclerosis.

We confirmed that the *APOE* ε3/ε3 genotype might increase ON development risk in males, while the ε3/ε4 genotype may reduce males’ risk of developing ON, as APOE serum levels were statistically significantly higher in the ON group than in the control group. However, because of a large number of SNPs, it is necessary to conduct replication studies in the future, particularly with a bigger sample size, to confirm the association between SNPs and serum levels and ON development.

## 5. Conclusions

The *APOE* ε3/ε3 genotype may increase ON development risk in males, while the ε3/ε4 genotype may reduce males’ risk of developing ON. APOE serum levels were statistically significantly higher in the ON group than in the control group.

## Figures and Tables

**Figure 1 genes-13-01188-f001:**
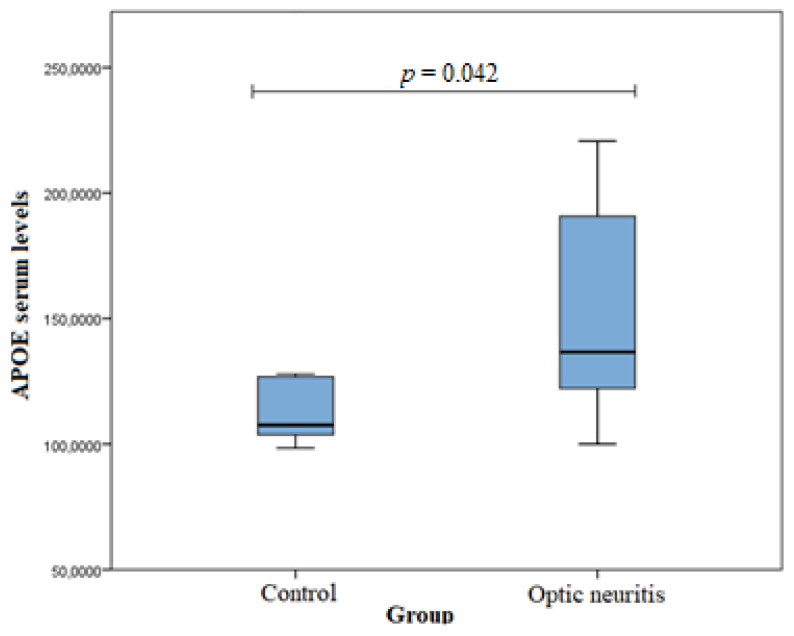
APOE Serum levels in control and ON groups.

**Table 1 genes-13-01188-t001:** Demographic characteristics of patients with optic neuritis (ON) and the control group subjects.

Characteristics	ON Group (*n* = 87)	Control Group (*n* = 583)	*p*-Value
Gender Females, *n* (%) Males, *n* (%)	56 (64.4) 31 (35.6)	276 (47.3) 307 (35.6)	0.003 *
Age, years (Median, IQR)	33 (16)	58 (34)	<0.001 *
Multiple sclerosis With Without Not determined	30 (34.5) 46 (52.9) 11 (12.6)		

* Mann–Whitney U test.

**Table 2 genes-13-01188-t002:** *APOE* genotype and allele frequencies in patients with ON and control groups.

Genotype, Allele	ON (*n* = 87)	Control (*n* = 583)	*p*-Value
*APOE* ε2/ε2, *n* (%)	0 (0)	3 (0.5)	-
*APOE* ε2/ε3, *n* (%)	14 (16.1)	84 (14.4)	0.774
*APOE* ε2/ε4, *n* (%)	1 (1.1)	17 (2.9)	0.342
*APOE* ε3/ε3, *n* (%)	57 (65.5)	387 (66.4)	0.874
*APOE* ε3/ε4, *n* (%)	15 (17.2)	85 (14.6)	0.516
*APOE* ε4/ε4, *n* (%)	0 (0)	7 (1.2)	-
ε2 allele, *n* (%)	15 (8.6)	107 (9.2)	0.812
ε3 allele, *n* (%)	143 (82.2)	943 (80.9)	0.681
ε4 allele, *n* (%)	16 (9.2)	116 (9.9)	0.756

**Table 3 genes-13-01188-t003:** APOE binary logistic regression in patients with ON and control groups.

Genotype	OR (95 %, CI)	*p*-Value
*APOE* ε2/ε2	-	-
*APOE* ε2/ε3	0.852 (0.440–1.652)	0.636
*APOE* ε2/ε4	1.285 (0.156–10.553)	0.815
*APOE* ε3/ε3	1.162 (0.669–1.930)	0.563
*APOE* ε3/ε4	0.765 (0.400–1.461)	0.416
*APOE* ε4/ε4	-	-

**Table 4 genes-13-01188-t004:** *APOE* genotype and allele frequencies in patients with ON and controls grouped by gender.

Genotype, Allele	Males		*p*-Value	Females		*p*-Value
	ON (*n* = 31)	Control (*n* = 307)		ON (*n* = 56)	Control (*n* = 276)	
*APOE* ε2/ε2, *n* (%)	0 (0)	1 (0.3)	-	0 (0)	1 (0.3)	-
*APOE* ε2/ε3, *n* (%)	7 (22.6)	48 (15.6)	0.318	7 (12.5)	48 (15.6)	0.547
*APOE* ε2/ε4, *n* (%)	1 (3.2)	8 (2.6)	0.838	0 (0)	8 (2.6)	-
*APOE* ε3/ε3, *n* (%)	15 (48.4)	204 (66.4)	0.045	42 (75.0)	204 (66.4)	0.208
*APOE* ε3/ε4, *n* (%)	8 (25.8)	43 (14.0)	0.080	7 (12.5)	43 (14.0)	0.764
*APOE* ε4/ε4, *n* (%)	0 (0)	3 (1.0)	-	0 (0)	3 (1.0)	-
ε2 allele, *n* (%)	8 (12.9)	58 (9.4)	0.382	7 (6.25)	58 (9.4)	0.276
ε3 allele, *n* (%)	45 (72.6)	499 (81.3)	0.100	98 (87.5)	499 (81.3)	0.113
ε4 allele, *n* (%)	9 (14.5)	57 (9.3)	0.186	7 (3.25)	57 (9.3)	0.298

**Table 5 genes-13-01188-t005:** *APOE* binary logistic regression in patients with ON and controls grouped by gender.

Group	Genotype	OR (95 %, CI)	*p*-Value
Males	*APOE* ε2/ε2	-	
	*APOE* ε2/ε3	0.540 (0.190–1,539)	0.249
	*APOE* ε2/ε4	0.258 (0.027–2.428)	0.236
	*APOE* ε3/ε3	3.698 (1.503–9.095)	0.004
	*APOE* ε3/ε4	0.242 (0.083–0.704)	0.009
	*APOE* ε4/ε4	-	-
Females	*APOE* ε2/ε2	-	-
	*APOE* ε2/ε3	1.125 (0.463–2.733)	0.794
	*APOE* ε2/ε4	-	-
	*APOE* ε3/ε3	0.644 (0.329–1.261)	0.200
	*APOE* ε3/ε4	1.359 (0.564–3.276)	0.495
	*APOE* ε4/ε4	-	-

**Table 6 genes-13-01188-t006:** *APOE* genotype and allele frequencies in patients with ON with MS and without MS.

Genotype, Allele	ON without MS (*n* = 46)	Control Group (*n* = 583)	*p*-Value	ON with MS (*n* = 30)	Control Group (*n* = 583)	*p*-Value
*APOE* ε2/ε2, *n* (%)	0 (0)	3 (0.5)	-	0 (0)	3 (0.5)	-
*APOE* ε2/ε3, *n* (%)	7 (15.2)	84 (14.4)	0.881	4 (13.3)	84 (14.4)	0.870
*APOE* ε2/ε4, *n* (%)	1 (2.2)	17 (2.9)	0.771	0 (0)	17 (2.9)	-
*APOE* ε3/ε3, *n* (%)	32 (69.6)	387 (66.4)	0.659	22 (73.3)	387 (66.4)	0.431
*APOE* ε3/ε4, *n* (%)	6 (13.0)	85 (14.6)	0.776	4 (13.3)	85 (14.6)	0.850
*APOE* ε4/ε4, *n* (%)	0 (0)	7 (1.2)	-	0 (0)	7 (1.2)	-
ε2 allele, *n* (%)	8 (8.7)	107 (9.2)	0.878	4 (7.1)	107 (9.2)	0.605
ε3 allele, *n* (%)	77 (83.7)	943 (80.9)	0.176	48 (85.8)	943 (80.9)	0.533
ε4 allele, *n* (%)	7 (7.6)	116 (9.9)	0.467	4 (7.1)	116 (9.9)	0.491

**Table 7 genes-13-01188-t007:** *APOE* binary logistic regression in patients with ON with MS and without MS.

ON Groups	Genotypes	OR (95 %, CI)	*p*-Value
ON with MS	*APOE* ε2/ε2	-	
	*APOE* ε2/ε3	1.065 (0.354–3.270)	0.910
	*APOE* ε2/ε4	-	-
	*APOE* ε3/ε3	0.781 (0.335–1.823)	0.568
	*APOE* ε3/ε4	1.055 (0.349–3.189)	0.925
	*APOE* ε4/ε4	-	-
ON without MS	*APOE* ε2/ε2	-	-
	*APOE* ε2/ε3	0.940 (0.392–2.255)	0.890
	*APOE* ε2/ε4	0.636 (0.067–5.353)	0.677
	*APOE* ε3/ε3	0.996 (0.505–1.967)	0.992
	*APOE* ε3/ε4	0.977 (0.387–2.467)	0.961
	*APOE* ε4/ε4	-	-

## Data Availability

Data will be provided if a request is made by editors, reviewers, or scientists.

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
