# Peer review of "Association of APOE Serum Levels and APOE ε2, ε3, and ε4 Alleles with Optic Neuritis"

_genes, 2022, doi:10.3390/genes13071188_

Round 1

Reviewer 1 Report

The manuscript presented by Liucija Momkute et al., needs some more extra work before it could be published. My main concern is regarding the flow, I found some difficulty to read the introduction and the discussion. I would suggest to the authors to rewrite the main concept in these paragraphs in order to improve the readability of this manuscript.

I have some comments\suggestions:

1.      In the introduction, page 2, line 58 the authors mentioned CNS with no the extensive name, please add it.

2.      Page 2 line 53-55 “Optic neuritis is considered to be a multifactorial demyelinating process. The causes of the disease at the genetic level are also examined. This article will investigate one of the blood serum proteins, Apolipoprotein E (ApoE).”

Regarding the introduction section I could not find the link between the ON and the ApoE, please rewrite better this part as is an essential point for all the manuscript.

3.      In the methods section the authors reported the group n as follow “The study participants were 89 subjects with optic neuritis and 69 healthy control subjects.” Why in all tables the ON group is considered with n=87 and control n=583? Please clarify this key point.

4.      Why the APOE serum was measured only in 20 subjects per group instead of all available subjects?

5.      Page 7 line 187- 191 “Serum ApoE levels were measured in the ON (n = 20) and healthy subject (n = 20) groups. Analysis was performed using the Mann-Whitney U test. A statistically significant difference was found between the groups (median (IQR): 136.65 (75.71) vs. 107.89 (25.14), p=0.045). ApoE serum levels were statistically significantly higher in the ON group than in the control group. The results are shown in figure 1.”

The p value reported in the figure 1 is 0.042 with no statistically significant symbol in. Is this correct?

Please recheck also this as I found not clear this point.

Author Response

Dear Editor and Reviewers,

We kindly appreciate the revision of our manuscript. We have highlighted the changes we made in the manuscript by using the track changes mode in MS Word. Hope that the revised manuscript will be acceptable for publication in your journal. Enclosed, please also find attached our point-by-point response to the comments raised by the reviewers (editors).

1st reviewer:

The manuscript presented by Liucija Momkute et al., needs some more extra work before it could be published. My main concern is regarding the flow, I found some difficulty to read the introduction and the discussion. I would suggest to the authors to rewrite the main concept in these paragraphs in order to improve the readability of this manuscript.

I have some comments\suggestions:

  1. In the introduction, page 2, line 58 the authors mentioned CNS with no the extensive name, please add it.

It was corrected.

  1. Page 2 line 53-55 “Optic neuritis is considered to be a multifactorial demyelinating process. The causes of the disease at the genetic level are also examined. This article will investigate one of the blood serum proteins, Apolipoprotein E (ApoE).”

Regarding the introduction section I could not find the link between the ON and the ApoE, please rewrite better this part as is an essential point for all the manuscript.

It was rewritten.

  1. In the methods section the authors reported the group n as follow “The study participants were 89 subjects with optic neuritis and 69 healthy control subjects.” Why in all tables the ON group is considered with n=87 and control n=583? Please clarify this key point.

Sorry, we increased the sample size. There should be 583 instead of 69 healthy controls. It was corrected.

  1. Why the APOE serum was measured only in 20 subjects per group instead of all available subjects?

Due to other publications, the ELISA sample size was performed only in 20 subjects per group. The analysis was performed twice regarding reliability.

  1. Page 7 line 187- 191 “Serum ApoE levels were measured in the ON (n = 20) and healthy subject (n = 20) groups. Analysis was performed using the Mann-Whitney U test. A statistically significant difference was found between the groups (median (IQR): 136.65 (75.71) vs. 107.89 (25.14), p=0.045). ApoE serum levels were statistically significantly higher in the ON group than in the control group. The results are shown in figure 1.”

The p value reported in the figure 1 is 0.042 with no statistically significant symbol in. Is this correct?Please recheck also this as I found not clear this point.

We confirm that 0,042 is statistically significant. We found that APOE serum levels were statistically significantly higher in ON than in the control group. Results were recalculated thus, 0,045 was not updated. Sorry for that.

Reviewer 2 Report

My suggestions:

1. Was there any other genetic risk factor described for Optic neuritis?

2. You may mention that APOE E2 allele could potentially protect against Alzheimer's disease onset.

3. A figure or table, of how different genotypes of APOE could potentially impact different disorders.

4. Were the APOE genotypes confirmed by Sanger sequencing? 

5. This study is a promising analysis of APOE genotypes in ON, and may be verified in larger amount of patients in the future. It would be interesting in the future to compare different populations/ethnic groups with ON with the current study. 

6. In discussion, a brief discussion may be useful, through what kind of mechanisms APOE E3/4 and E3/3 could increase the ON risk?

Author Response

Dear Editor and Reviewers,

We kindly appreciate the revision of our manuscript. We have highlighted the changes we made in the manuscript by using the track changes mode in MS Word. Hope that the revised manuscript will be acceptable for publication in your journal. Enclosed, please also find attached our point-by-point response to the comments raised by the reviewers (editors).

2nd reviewer:

  1. Was there any other genetic risk factor described for Optic neuritis?

No, there was not.

  1. You may mention that APOE E2 allele could potentially protect against Alzheimer’s disease onset.

It was mentioned (page 2, line 63).

  1. A figure or table, of how different genotypes of APOE could potentially impact different disorders.

The only disorder that is briefly investigated with APOE  is Alzheimer’s disease. There is no evidence yet that APOE plays a critical role in different disorders. The gene (genotypes) is involved in transporting fat molecules associated with LDL and VLDL, but there is a lack of evidence of disorders development.

  1. Were the APOE genotypes confirmed by Sanger sequencing?

No, it was selected based on a literature review. Both SNPs play a significant role in other disease but has not been investigated with ON.

  1. This study is a promising analysis of APOE genotypes in ON, and may be verified in larger amount of patients in the future. It would be interesting in the future to compare different populations/ethnic groups with ON with the current study.

We totally agree with you; we hope to increase the sample size in the future.

  1. In discussion, a brief discussion may be useful, through what kind of mechanisms APOE E3/4 and E3/3 could increase the ON risk?

The mechanisms are not well understood yet. We investigated that these SNPs impact ON development, but a deeper investigation is needed to understand molecular mechanisms briefly.

Round 2

Reviewer 1 Report

In the present form I think that the paper could be accepted for publication.

Reviewer 2 Report

The manuscript is acceptable now